# SFDA-CD: A Source-Free Unsupervised Domain Adaptation for VHR Image Change Detection

**Jingxuan Wang and Chen Wu *** ⓘ

State Key Laboratory of Information Engineering in Surveying, Mapping and Remote Sensing, Wuhan University, Wuhan 430072, China; raintohsaka@whu.edu.cn
* Correspondence: chen.wu@whu.edu.cn

**Abstract:** Deep models may have disappointing performance in real applications due to the domain shifts in data distributions between the source and target domain. Although a few unsupervised domain adaptation methods have been proposed to make the pre-train models effective on target domain datasets, constraints like data privacy, security, and transmission limits restrict access to VHR remote sensing images, making existing unsupervised domain adaptation methods almost ineffective in specific change detection areas. Therefore, we propose a source-free unsupervised domain adaptation change detection structure to complete specific change detection tasks, using only the pre-trained source model and unlabelled target data. The GAN-based source generation component is designed to generate synthetic source data, which, to some extent, reflects the distribution of the source domain. Moreover, these data can be utilised in model knowledge transfer. The model adaptation component facilitates knowledge transfer between models by minimising the differences between deep features, using AAM (Attention Adaptation Module) to extract the difference between high-level features, meanwhile we proposed ISM (Intra-domain Self-supervised Module) to train target model in a self-supervised strategy in order to improve the knowledge adaptation. Our SFDA-CD framework demonstrates superior accuracy over existing unsupervised domain adaptation change detection methods, which has 0.6% cIoU and 1.5% F1 score up in cross-regional tasks and 1.4% cIoU and 1.9% F1 score up in cross-scenario tasks, proving that it can effectively reduce the domain shift between the source and target domains even without access to source data. Additionally, it can facilitate knowledge transfer from the source model to the target model.

**Keywords:** change detection; unsupervised domain adaptation; source-free; deep learning

## 1. Introduction

Very-high-resolution (VHR) Image Change Detection refers to the process of identifying and analysing changes in objects or phenomena within a particular area using two VHR images captured at different times. It has been widely used in numerous practical applications, such as environmental monitoring [1,2], disaster assessment [3], land cover and land use status [4,5], urban expansion [6], etc.

Recent research has extensively explored deep learning-based methods, for example, deep neural networks, as a fundamental VHR image change detection technology. These methods possess potent features that can effectively learn from and extract features in complex scenarios. However, the successful application of deep models relies on producing large-scale, densely labelled remote-sensing image datasets [7,8], which can be cost- and labor-expensive. An intuitive solution is transferring the knowledge from existing well-trained models on source datasets to unlabeled target domains, called Domain Adaptation [9]. Yet, it still faces the challenge of domain shifts in data distributions between the source and target domain [10], which can be demonstrated below:

$$D_{DomainShift} = \alpha \| P_S(x,y), P_T(x,y) \| + \beta \| D_S, D_T \| \tag{1}$$

while $D_S = \{(x_i^S, y_i^S)\}_{i=1}^m, D_T = \{(x_i^T, y_i^T)\}_{i=1}^n$ are source and target domain datasets, $P_S(x, y), P_T(x, y)$ are probability distributions of $D_S$ and $D_T$, $\|\|$ means specific norm calculation. Meanwhile, it also requires a portion of labelled target data, although the proportion of labelled data in the target data can be deficient.

Unsupervised domain adaptation (UDA) has been proposed to address these issues [11]. Unlike traditional domain adaptation methods, UDA methods can generate a well-trained model on an unlabeled target domain, effectively avoiding the expensive data annotation process. Yet these methods still rely on well-trained source models and fully labelled source data, as source data plays a vital role in the domain adaptation process [12]. It helps maintain the knowledge from the source domain during the training of the target model, which is essential for reducing cross-domain discrepancy. However, due to the geo-information in VHR remote sensing images, constraints like data privacy, security, and transmission limits [13,14] restrict access to source datasets in specific VHR change detection areas; only well-trained source models and unlabeled target data are available. In such scenarios, existing unsupervised domain adaptation methods are almost ineffective because of the unavailability of source data and the low credibility of target domain supervision information.

With the above insights, a new source-free unsupervised domain adaptation method has been proposed. As shown in Table 1, this method requires only the well-trained source model and unlabeled target domain data to complete the domain adaptation process. Recently, from 2021, a few source-free UDA methods have been developed to address similar issues, such as scene classification [15]. However, deep learning-based tasks for VHR image change detection are pixel-level and fundamentally different from image-level tasks like scene classification. Pixel-level change detection tasks require identifying the semantics of each pixel at the same location on an image pair—changed or unchanged—and these semantic features are extracted during the model's encoding process. Considering that the knowledge of these features cannot be utilized without source data, we attempt to use the available source model, using partial statistic information recorded in the source model [16] to generate source-like image data. This approach, called domain generation, helps to some extent in recovering and transferring the knowledge learned by the source model.

**Table 1.** Contents Accessibility of Different Domain Adaptation Tasks.

| | Traditional Domain Adaptation | Unsupervised Domain Adaptation | Source-Free Unsupervised Domain Adaptation |
|---|---|---|---|
| Source Model | ✓ | ✓ | ✓ |
| Source Data | ✓ | ✓ | ✗ |
| Target Data | Part-labelled | No label | No label |

In this work, we propose a framework named Source-Free Domain Adaptation Change Detection (SFDA-CD) for VHR image change detection for the first time. This framework first extracts information about the source domain from a well-trained source model. Meanwhile, based on domain generation, this information guides the independent, trainable generator to synthesise a set of fake samples highly similar to the source dataset in data distribution. These fake samples can be considered the source dataset and used to transfer knowledge between the source and target models. By combining the "source data" with the unlabeled target data, we transform it into a UDA issue. Moreover, the core of the deep change detection networks lies in identifying and extracting change features, and the attention mechanism has been proposed to solve the loss of model performance caused by indistinct distributed changed features in VHR image change detection tasks. Therefore, a dual-attention mechanism is introduced to allow the generator to notice valuable feature information and make the fake samples closer to the source dataset. At the same time, it also enables the target model to pay attention to valuable knowledge during knowledge

transfer. We also employ an intra-domain adaptation self-supervised module to preserve the credibility of supervision information on the target domain due to the lack of fully annotated target data. Our main contributions can be summarised as follows:

- We propose a domain generation-based SFDA framework for change detection. This framework can adapt from the source domain to the target domain without any source data and target data annotation, which is essential yet rare in real-world tasks;
- We utilise domain generation methods to synthesise fake samples, addressing the lack of source data. We employ a dual-attention mechanism to ensure the framework captures valuable changed semantic information during training. Meanwhile, we adopt an intra-domain adaptation self-supervised module to obtain more accurate detection maps for self-supervision;
- We demonstrate the efficiency of this framework in cross-regional and cross-scenario change detection tasks. It achieves accuracy comparable to current state-of-the-art source-driven UDA methods, in cross-regional tasks our method has 0.6% cIoU and 1.5% F1 score up, and in cross-scenario tasks has 1.4% cIoU and 1.9% F1 score up. Both qualitative results have shown that our methods can effectively avoid inner hole and detect edge precisely.

## 2. Related Works

### 2.1. Fully Convolutional Networks-Based VHR Image Change Detection Frameworks

Most deep VHR-CD frameworks are based on a U-net structure, which features a U-shaped encoder-decoder design, skip-connection, and a fully convolutional backbone network. U-net [17] was first used for medical image segmentation in 2015. In 2018 Daudt [18] later proposed a fully convolutional change detection network based on the U-net structure, utilising an Early-fusion and Siamese-encoder design. The U-net framework is highly flexible; some studies [19,20] achieve better encoding effects by replacing the backbone network with extensive networks like the VGG and ResNet series between 2016 and 2018. Some researchers have focused on the skip-connection between the encoder and decoder [21,22], believing skip-connections can transfer features at different depths. This prevents the loss of detail caused by the low resolution of deep semantic features during decoding. In 2020, Introducing attention mechanisms [23,24] solved the issue of changed semantic feature distribution in change detection tasks. Spatial and channel attention mechanisms [25,26] allow the networks to focus more on the associations and contextual information. This avoids training difficulties and overfitting caused by the unbalanced distributions between changed and non-changed classes. Recently, some scholars have utilized the feature of Transformers to extract global features, combining convolutional encoding with Transformers to fuse multi-scale global-local features and enhance change detection accuracy [27]. Li et al. [28] proposed TMM based on the self-attention mechanism of Transformers to encode multi-level features, enhancing the Transformer's ability to process multi-temporal data while reducing computational complexity.

### 2.2. Unsupervised Domain Adaptation for VHR Image Change Detection

Methods for VHR Image Change Detection using Unsupervised Domain Adaptation (UDA) can be classified into several categories. Some UDA methods rely on adversarial or contrastive learning [29] to reduce cross-domain discrepancy in 2020, focusing on aligning the distributions between the source and target domain. Several ways use generative models such as image-to-image translation or Generative Adversarial Networks (GANs) [30–32] to generate target data based on the source data features, narrowing the differences like illumination and colour between the source and target domain. Moreover, since UDA methods are applied to unlabeled target domains, one of the critical challenges is addressing supervision information in the target domain. From 2021 some proposed approaches employ self-supervision [33,34], generating pseudo-labels to provide simple supervision information. Wang et al., proposed utilizing Markov Random Fields to conduct change detection on multi-source heterogeneous remote sensing images (primarily optical and

SAR images) over long time series [35]. However, all the UDA methods mentioned above require access to fully annotated source data, which could be unavailable in real-world scenarios due to data privacy, security, and transmission limits.

*2.3. Source-Free Unsupervised Domain Adaptation Based on Domain Generation*

Domain generation was first proposed as a solution to the problem of limited data access [36] in 2021. This aligns perfectly with the application scenarios of source-free UDA. There are currently two types of domain generation methods: domain image generation and domain distribution generation. In domain image generation, from 2022, some studies focus on the Batch Normalisation layers of the source model, which store the mean and variance of each mini-batch during the model's training. These studies utilise BN (Batch Normalization) statistical information for image style transfer [37]. Other research has trained generators based on GANs (Generative Adversarial Networks) and then used Knowledge Distillation (KD) modules to extract knowledge from the source model [38]. This is done to generate images that reflect the style of the source domain [39] or to transfer the style of target domain images to the source domain. In domain distribution generation, studies assume that data within a part conforms to a specific distribution. In 2022 Some methods use Gaussian Mixture Models (GMM) [40] to directly simulate the source domain data distribution and combine it with adversarial training techniques to minimise the discrepancy between the simulated source domain and the target domain. Since VHR-CD is pixel-level, some methods applicable to image-level tasks could be more suitable. Finally, we chose a domain image generation method based on GAN generators and statistical information to simulate source data.

## 3. Methodology

Existing source-driven UDA methods define a fully annotated source dataset $D_{\mathcal{S}}$, an unlabeled target domain dataset $D_{\mathcal{T}}$, and a well-trained source model $\mathcal{S}$ to sufficiently train the target model $\mathcal{T}$, which shares parameters with $\mathcal{S}$, using the target domain dataset $D_{\mathcal{T}}$. Therefore, source-driven UDA methods can be formalised as follows:

$$\mathcal{L}_{UDA} = \mathcal{L}_{Src}(\mathcal{S}, D_{\mathcal{S}}) + \mathcal{L}_{Tar}(D_{\mathcal{T}}) \tag{2}$$

$\mathcal{L}_{Src}$ is a supervised loss used to maintain and transfer knowledge of the source domain; $\mathcal{L}_{Tar}$ is a self-supervised loss based on pseudo-labels to measure the performance of the target model in the target domain, such as entropy loss [41], maximum square loss [42], etc.

Using source domain knowledge for supervised adaptation is impossible without labelled source data, which is the most challenging task for the source-free scenario. However, we can assume that certain parts of the source model reflect the features of the source domain based on the source domain information preserved in the model. Since the source domain performs well only on source data, we can estimate the features of the source data and transfer the knowledge from the estimated source data to the target domain during the adaptation process.

Our Source-free Unsupervised Domain Adaptation framework comprises two main components: source generation and model adaptation, which are based on certain principles. Figure 1 illustrates the overall structure of the framework. Based on GAN theory, we utilise a generator $\mathcal{G}_1$ and $\mathcal{G}_2$ to generate specific data $\{\tilde{x}_{\mathcal{S}}^1, \tilde{x}_{\mathcal{S}}^2\}$ by inputting random noise in the source generation component. The generator is linked to the source model $\mathcal{S}$ at the end. To ensure that the source domain knowledge can be effectively extracted and transferred, we replicate another source model $\mathcal{S}'$ and fix its parameters throughout the training process. We introduce an attention adaptation mechanism between the source models. This mechanism focuses on the differences in the features encoded by the two source models, thereby constraining the generator. In the model adaptation component, we introduce a structure consistent with the source model but entirely initialised as the target model $\mathcal{T}$. The end of the target model is linked to an intra-domain adaptation

self-supervision module, which maximises the usage of correctly discriminated parts in pseudo-labels to improve the usability of the target data $\{x_{\mathcal{T}}^1, x_{\mathcal{T}}^2\}$. Hence, the whole source-free UDA framework can be formalised as below:

$$\mathcal{L}_{SF-UDA} = \alpha\mathcal{L}_G\left(\tilde{x}_{\mathcal{S}}^1, \tilde{x}_{\mathcal{S}}^2\right) + \beta\mathcal{L}_{MA}\left(\tilde{x}_{\mathcal{S}}^1, \tilde{x}_{\mathcal{S}}^2, x_{\mathcal{T}}^1, x_{\mathcal{T}}^2\right) \tag{3}$$

We have devised an optimization problem to abstract the entire framework, which comprises two main components. The first component is $\mathcal{L}_G$, which is a hybrid loss function used to restrict the part of the framework responsible for source generation. The second component is $\mathcal{L}_{MA}$, which is an unsupervised loss function. It offers constraints at various points within the framework to facilitate the model adaptation process. The SFDA-CD structure is described as the following pseudocoder (Algorithm 1):

---

**Algorithm 1** SFDA-CD

---

**Input:** Random gaussian noise $z_1$, $z_2$, bi-temporal VHR image pairs $\{x_{\mathcal{T}}^1, x_{\mathcal{T}}^2\}$
1: **Initialization:** Generators $\mathcal{G}_1, \mathcal{G}_2$, target model $\mathcal{T}$, hyper-parameters: epoch $n \leftarrow 0$, $\alpha = 0.5$, $\beta = 0.5$;
2: **Freeze:** Source model $\mathcal{S}$ (except BN layers), source model $\mathcal{S}'$;
3: **for** epoch $n = 1$ to Max Epoch: $N$ **do**
4:     **Forward**
5:       **Source Generation:** $\tilde{x}_{\mathcal{S}}^1, \tilde{x}_{\mathcal{S}}^2 = \mathcal{G}_1(z_1), \mathcal{G}_2(z_2)$;
6:       $\mathcal{L}_{SF-UDA} = \alpha\mathcal{L}_G\left(\tilde{x}_{\mathcal{S}}^1, \tilde{x}_{\mathcal{S}}^2\right) + \beta\mathcal{L}_{MA}\left(\tilde{x}_{\mathcal{S}}^1, \tilde{x}_{\mathcal{S}}^2, x_{\mathcal{T}}^1, x_{\mathcal{T}}^2\right)$
7:     **Backward**
8:       Update parameters in $\mathcal{T}$;
9:     **until** $\mathcal{L}_{SF-UDA} < \epsilon$
10: **end for**
11: Update parameters in $\mathcal{T}$;
**Output:** Change map $x_{CD} = \mathcal{T}\left(x_{\mathcal{T}}^1, x_{\mathcal{T}}^2\right)$

---

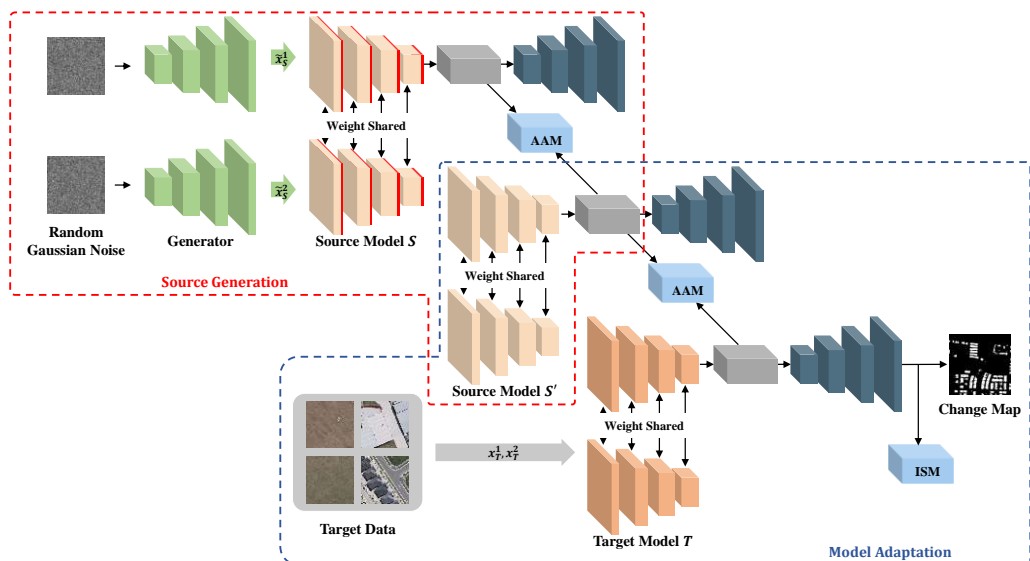

**Figure 1.** Architecture of proposed SF-UDA CD framework.

### 3.1. Source Generation

Pixel-level change detection tasks that rely on deep learning have a specific requirement. They need to process two images simultaneously, and most datasets have two sets of images from different time phases. To adapt the model, the domain generation process must create two sets of composite fake samples with specific differences. These generated fake samples will act as the "source data" and contain source domain features.

Thankfully, most of the mainstream fully convolutional change detection frameworks use a Siamese encoder structure. Each encoder corresponds to an input image from the one-time phase, and the Siamese encoder structure can identify differences between the two inputs while recording the differences in the input data during training. The Batch Normalisation Statistics (BNS) [43] in each encoder provide this difference, which is crucial for generating the domain and creating two sets of differentiated fake samples. Hence, we designed two generators $\mathcal{G}_1$ and $\mathcal{G}_2$ to generate the unobtainable source data $\{\tilde{x}_S^1, \tilde{x}_S^2\}$. Due to the balance of source generation's performance with the framework's overall training difficulty, we used a generator structure similar to [44]. The details of source generation is shown in Figure 2:

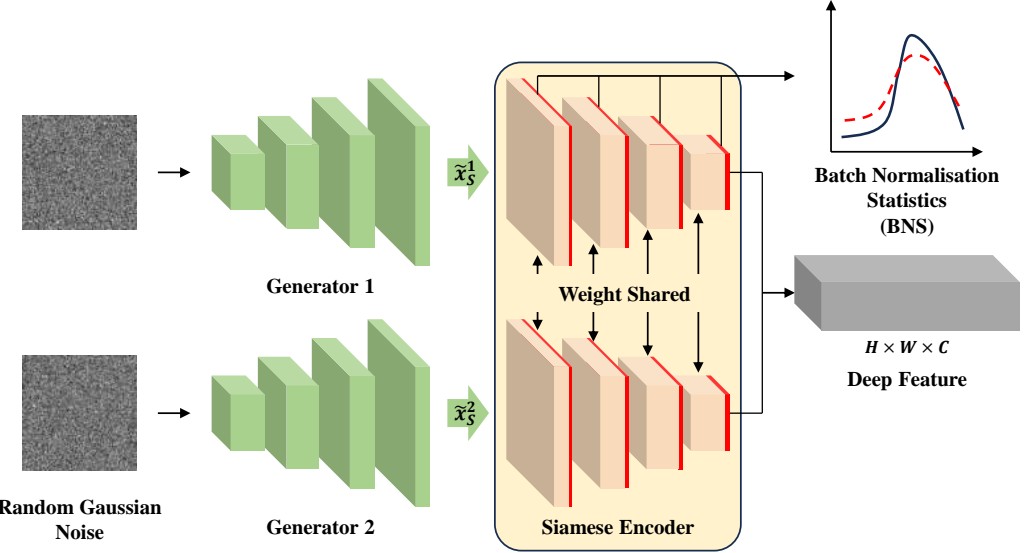

**Figure 2.** Source Generator of Siamese Change Detection Encoder.

The input to generators is a set of random noises $z$ that follow the Gaussian distribution.

$$\tilde{x}_S^i = \mathcal{G}^i(z), z \sim N(0,1), i \in 1,2 \tag{4}$$

Once the synthetic data $\tilde{x}_S^i$ have been generated, they are processed by the source models $\mathcal{S}$ and $\mathcal{S}'$ as if they were "real" bi-temporal remote sensing images. The generators $\mathcal{G}_1$ and $\mathcal{G}_2$, meanwhile, operate under the constraints of the BNS provided by the source model $\mathcal{S}$ during the training process.

In the traditional GAN architecture, the loss function for the generator is typically described as follows:

$$\mathcal{L}_G = -\mathbb{E}_{z \sim p_z(z)}[logD(\mathcal{G}(z))] \tag{5}$$

where $D(\mathcal{G}(z))$ represents the discriminator's judgment of the data generated by the generator. However, in our framework, the two generators correspond to two encoders in the source model. The BNS in different encoders impose constraints on the respective generators. We unfreeze the BN layers during training in model $\mathcal{S}$. The reason behind this is that the synthesised images in each epoch during domain generation show changes in mean and standard deviation, as recorded by the BN layers of the model. Our optimisation objective is to minimise the discrepancy with the BNS recorded in the original source model $\mathcal{S}'$ as much as possible. Therefore, the loss function can be reformulated as follows:

$$\mathcal{L}_G = \frac{1}{2}[\sum_{i=1}^{2} \mathcal{L}_{BNS}\mathcal{G}_i(z_i)] + \mathcal{L}_{AAM}^{\mathcal{SS}'} \tag{6}$$

$\mathcal{L}_{BNS}$ ensures that the generators produce high-reliability synthesis image pairs by extracting the corresponding positions of BNS between the source models $\mathcal{S}$ and $\mathcal{S}'$ and computing the differences between them. Meanwhile, $\mathcal{L}_{AAM}^{\mathcal{SS}'}$ measures the differences between the attention maps of $\mathcal{S}$ and $\mathcal{S}'$. Detailed definitions of each loss function will be introduced in subsequent sections.

### 3.2. Attention Adaptation Module

Change detection tasks often face an imbalance in quantity and distribution between the classes that have changed and those that have not. This can lead to deep models failing to focus on changed features as much as they should. Meanwhile, in domain adaptation, the goal is to enable the target model to focus on features from the source domain that are most relevant to the target domain and reduce the distributional differences in those features between the two domains. Our framework aims to tackle model adaptation tasks by reducing the feature-level discrepancies between the source and target models. We also aim to regulate the quality of synthetic sample generation during training. We propose an Attention Adaptation Module (AAM) to achieve these goals, as illustrated in Figures 3 and 4.

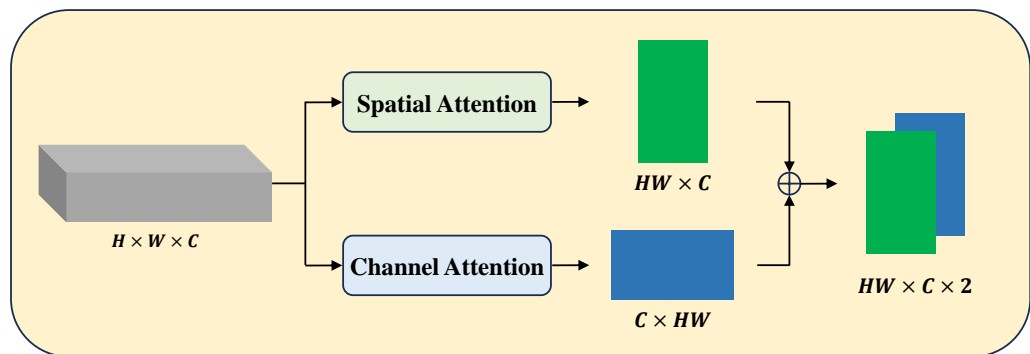

**Figure 3.** The illustration of AAM.

The feature map extracted by the source change detection model is denoted as $F^{H \times W \times C} = \mathcal{F}(\tilde{x}_S^1, \tilde{x}_S^2)$. $\mathcal{F}$ represents the encoder function of source model $\mathcal{S}$. It is noted that $H$, $W$, and $C$ represent height, width, and the number of channels. AAM comprises spatial and channel attention mechanisms, as illustrated in Figure 4. The feature map $F^{H \times W \times C}$ is first reshaped into $F^{HW \times C}$. Subsequently, the spatial attention map $S \in \mathbb{R}^{HW \times HW}$ is computed by:

$$s_{ij} = \frac{exp\left(F_{[i,:]} \cdot F_{[:,j]}^{\top}\right)}{\sum_{i}^{HW} exp\left(F_{[i,:]} \cdot F_{[:,j]}^{\top}\right)} \tag{7}$$

where $F^{\top}$ is the transpose of $F^{HW \times C}$, and $s_{ij}$ shows the influence between the pixel at position $i$ and the pixel at position $j$. Simultaneously, the channel attention map $C \in \mathbb{R}^{C \times C}$ is calculated by:

$$c_{ij} = \frac{exp\left(F_{[i,:]} \cdot F_{[:,j]}^{\top}\right)}{\sum_{i}^{C} exp\left(F_{[i,:]} \cdot F_{[:,j]}^{\top}\right)} \tag{8}$$

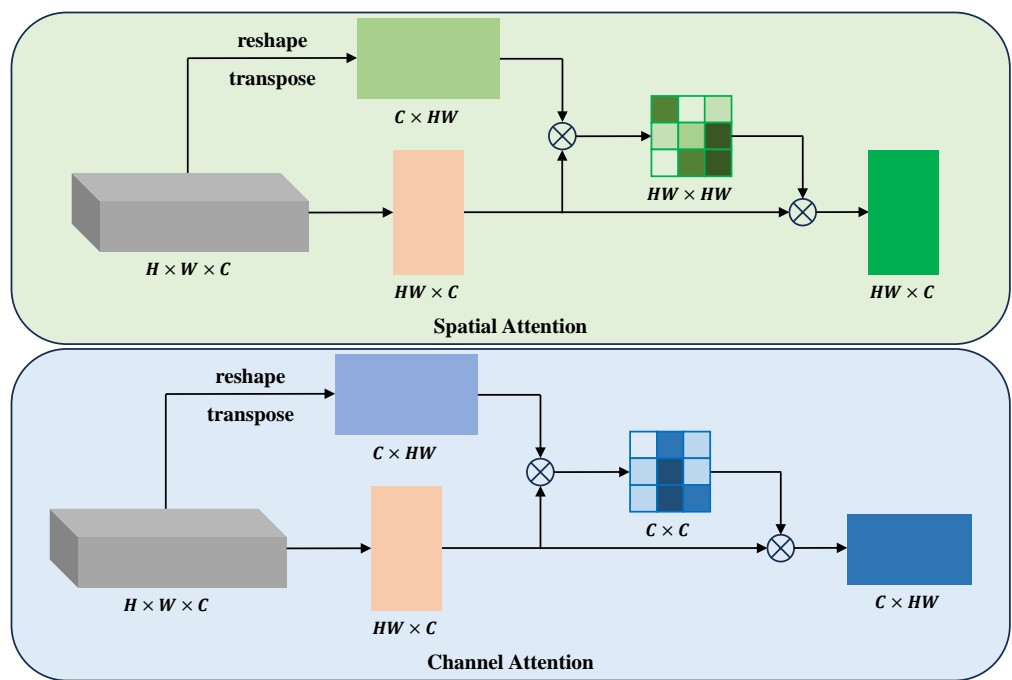

**Figure 4.** The Details of AAM, including spatial and channel attention mechanism.

Our approach differs from traditional attention modules in that we do not superimpose the attention maps onto the original features for feature fusion. Instead, we concentrate solely on the attention maps derived from these features and their differences. We use these differences to restrict source generation and model adaptation. The constraint mechanism of AAM in source generation can be described as follows:

$$\mathcal{L}_{AAM}^{\mathcal{SS}'} = \mathbb{E}_{\tilde{x}_s}(\|\mathcal{M}(\bar{\mathcal{F}}(\tilde{x}_\mathcal{S}^1, \tilde{x}_\mathcal{S}^2)) - \mathcal{M}(\mathcal{F}(\tilde{x}_\mathcal{S}^1, \tilde{x}_\mathcal{S}^2))\|) \tag{9}$$

In model adaptation, the constraints can be denoted as:

$$\mathcal{L}_{AAM}^{\mathcal{S}'\mathcal{T}} = \mathbb{E}_{\tilde{x}_s}(\|\mathcal{M}(\bar{\mathcal{F}}(\tilde{x}_\mathcal{S}^1, \tilde{x}_\mathcal{S}^2)) - \mathcal{M}(\tilde{\mathcal{F}}(x_\mathcal{T}^1, x_\mathcal{T}^2))\|) \tag{10}$$

where $\mathcal{M} = concat(R \bullet F | F \bullet S)$ is the concatenation of spatial and channel maps and $\tilde{F}^{H \times W \times C} = \tilde{\mathcal{F}}(x_T^1, x_T^2)$ is the feature map extracted by target model. Details such as the selection and calculation of norms will be elucidated in subsequent sections.

### 3.3. Intra-Domain Self-Supervision Module

Given the absence of labels in the target data, the training of the target model can be considered an unsupervised or self-supervised process. This implies the necessity for pseudo-labels derived from the target data. We observe that the target model can predict with reasonable accuracy in certain areas, and these accurately predicted pseudo-labels provide effective supervisory information for model adaptation and training. In CVPR 2020, Pan [45] proposed an unsupervised approach for joint inter-domain and intra-domain adaptation, which categorises the target domain based on the difficulty of model prediction using an entropy-ranking method. Furthermore, this approach utilises adversarial learning mechanisms to minimise inter- and intra-domain discrepancies.

We introduced an Intra-domain Self-supervision Module to enhance the previously mentioned concept, as shown in Figure 5. While operating, this module gathers feature maps of all target data in each mini-batch and calculates their entropy maps. During training, change detection models usually resize the dataset to smaller-sized images and filter out areas with no change to maintain a balance between classes. This technique enables the model to train with a larger batch size under specific computational constraints.

We utilise this method to acquire features and compute entropy within a mini-batch. Subsequently, based on these calculations, the data is categorised into two parts: high credibility and low credibility. Finally, a specifically designed discriminator evaluates these data segments, and an adversarial loss function is utilised to enforce constraints.

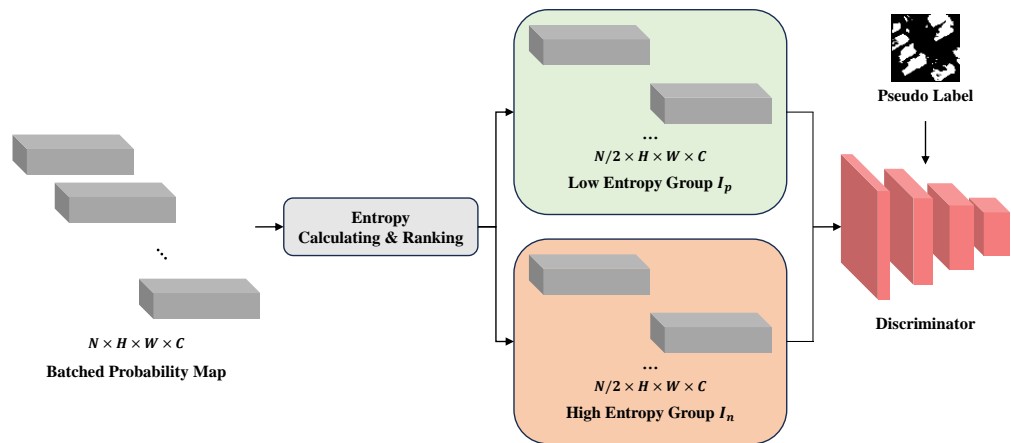

**Figure 5.** The illustration of ISM.

In this module, we define the size of each batch as N, and the whole batch can be denoted as $\mathcal{N} = x_i, i \in [1, N], i \in \mathbb{N}$, with $x_i, i \in [1, n], i \in \mathbb{N}$ representing each data pair in the target domain. We noted that all the images in the target domain have the same size as $H \times W$. At the end of the decoder in the target model, a softmax function is used to create probability maps $p_i$. We then calculate the entropy for each probability map as below:

$$E_{x_i} = -\frac{1}{H \times W} \sum_{h \times w}^{H \times W} \sum_c^C p_i^{h \times w \times c} log\left(p_i^{h \times w \times c}\right) \tag{11}$$

Then we rank each probability map $p_i$ and its corresponding target data $x_i$ based on the entropy map $E_{x_i}$. We then divide these data into two groups: a group of probability maps with lower entropy $I_p$, considered to have higher credibility. We can be used as pseudo-labels and a group with higher entropy $I_n$ as the below process:

$$I_p, I_n \leftarrow Rank(E_{x_i}|x_i \in \mathcal{N}) \tag{12}$$

A discriminator $\mathcal{D}$ is designed to determine whether the input probability maps belong to group $I_p$ or $I_n$. The discriminator $\mathcal{D}$'s structure still follows the classic structure proposed by [44]. However, the input channel size has been modified to accommodate the target model's prediction map channel size. This modification is based on the principle of balancing discrimination quality and model training efficiency, similar to the selection principle of the generator structure. The probability maps generated by the target model should be challenging for $\mathcal{D}$, reducing the differences between group $I_p$ and $I_n$. This aims to improve the overall quality of pseudo-labels within a mini-batch. The adversarial learning loss is designed as follows:

$$\mathcal{L}_{ISM} = -\sum_{i=1}^n (\sum_p^P log\left(1 - \mathcal{D}\left(i, E_{x_i}^p\right)\right) + \sum_n^{N-P} log\left(\mathcal{D}\left(E_{x_i}^n\right)\right)) \tag{13}$$

*3.4. Loss Function*

Based on (1) and the framework definitions, we can write the overall loss function:

$$\mathcal{L}_{SF-UDA} = \min_{\mathcal{S}, \mathcal{S'}} \alpha \left(\frac{1}{2}\left[\sum_{i=1}^2 \mathcal{L}_{BNS} G^i(z)\right] + \mathcal{L}_{AAM}^{\mathcal{SS'}}\right) + \min_{\mathcal{T}, \mathcal{S}} \max_{\mathcal{D}} \beta\left(\mathcal{L}_{AAM}^{\mathcal{S'T}} + \mathcal{L}_{ISM} + \mathcal{L}_{TAR}\right) \tag{14}$$

$\mathcal{L}_{BNS}$ is defined based on the differences in the BNS of the source model $\mathcal{S}$ and $\mathcal{S}$ before and after receiving the generated source data:

$$\mathcal{L}_{BNS} = \sum_i^L \left( \|\mu_i(\tilde{x}_{\mathcal{S}}^1, \tilde{x}_{\mathcal{S}}^2) - \bar{\mu}_i\|_2^2 + \|\sigma_i(\tilde{x}_{\mathcal{S}}^1, \tilde{x}_{\mathcal{S}}^2) - \bar{\sigma}_i\|_2^2 \right) \tag{15}$$

In this loss function, the variables $\bar{\mu}_i$ and $\bar{\sigma}_i$ correspond to the mean and standard deviation values stored in the BN layers of the source model $S'$ which is completely frozen, and these parameters represent the statistical characteristics of the source domain data at each stage when encoded through the source model.

Both $\mathcal{L}_{AAM}^{\mathcal{SS}'}$ and $\mathcal{L}_{AAM}^{\mathcal{S}'\mathcal{T}}$ (as presented in (8) and (9), respectively) have a similar structure but differ in their application of norms. When calculating the $\mathcal{L}_{AAM}^{\mathcal{SS}'}$ between $\mathcal{S}$ and $\mathcal{S}'$, the 1-norm is used, The source models $\mathcal{S}$ and $\mathcal{S}'$ have nearly identical structures, except for the BN layers adjusted during training. All other layers in both models have the same parameters, and the input data used for training is synthetic data generated in the same epoch. This ensures no complex differences between the deep features encoded by the two models and the attention maps processed by AAM involving subspaces. Therefore, a simple 1-norm can achieve the constraint effect while reducing computational demands. while for $\mathcal{S}'$ and $\mathcal{T}$, the Kullback–Leibler (KL) divergence is computed separately on the spatial and channel attention maps, This is because although the target model $\mathcal{T}$ and the source model $\mathcal{S}'$ have the same network structure, they differ in the input data and internal network parameters, resulting in significant differences between the features extracted by each. To minimise the loss incurred when approximating target feature distribution with source feature distribution, we use the KL divergence measure. In domain adaptation, the goal is to align the features extracted by the target model as closely as possible with those of the source model. Using KL divergence as a loss function, we can constrain the loss during alignment, enabling the target model to extract features similar to the source model. Thus, completing the domain adaptation process:

$$\mathcal{L}_{AAM}^{\mathcal{SS}'} = \mathbb{E}_{\tilde{x}_s} \left( \frac{1}{HW \times C \times 2} \|\mathcal{M}(\tilde{F}(\tilde{x}_{\mathcal{S}}^1, \tilde{x}_{\mathcal{S}}^2)) - \mathcal{M}(F(\tilde{x}_{\mathcal{S}}^1, \tilde{x}_{\mathcal{S}}^2))\|_1 \right) \tag{16}$$

$$\mathcal{L}_{AAM}^{\mathcal{S}'\mathcal{T}} = \mathbb{E}_{\tilde{x}_s} \left( \sum_i^2 D_{KL}\left( \mathcal{M}^{HW \times C \times i}\left( \mathcal{F}\left(\tilde{x}_{\mathcal{S}}^1, \tilde{x}_{\mathcal{S}}^2\right) \right), \mathcal{M}^{HW \times C \times i}\left( \mathcal{G}\left(x_{\mathcal{T}}^1, x_{\mathcal{T}}^2\right) \right) \right) \right) \tag{17}$$

The $\mathcal{L}_{TAR}$ is an entropy loss function that restricts the target model:

$$\mathcal{L}_{TAR} = -\frac{1}{log(C)} \sum_{h,w}^{H,W} \sum_c^C p_t^{h,w,c} log\left( p_t^{h,w,c} \right) \tag{18}$$

Probability value $p_t^{h,w,c}$ at position (h,w,c) on target data's probability map.

## 4. Experiment

### 4.1. Experiment Settings

#### 4.1.1. Datasets

We set up two experiments to evaluate the proposed SFDA-CD framework: cross-regional and cross-scenario, and we chose different datasets for each experiment.

In the cross-regional experiment, following the previous work, we configured the WHU-CD [46] and LEVIR-CD [24] as the datasets; each dataset will be used as the source domain and the target domain. WHU-CD dataset covers an area where a 6.3-magnitude earthquake occurred in February 2011 and was rebuilt in the following years. This dataset consists of aerial images obtained in April 2012 that contain 12,796 buildings in 20.5 km$^2$ (16,077 buildings in the same area in the 2016 dataset). Both images are 32,207 × 15,354 pixels, with a spatial resolution of 0.2 m. LEVIR-CD is a large-scale re-

mote sensing Building Change Detection dataset widely used in the benchmark of deep learning-based change detection algorithms. It consists of 637 very high-resolution (VHR, 0.5 m/pixel) Google Earth (GE) image patch pairs with a size of 1024 × 1024 pixels. The fully annotated LEVIR-CD contains a total of 31,333 individual change-building instances.

In the cross-scenario experiment, we set the CDD [47] as the source domain and the WHU-CD and LEVIR-CD as the target domains. The CDD dataset contained 16,000 image sets with 256 × 256 pixels: 10,000 train sets and 3000 test and validation sets, which obtained seasonal images for manual ground truth creation and minimal change images for manually adding objects. The spatial resolution of these images was from 3 to 100 cm/pixel. This dataset provided various scenarios of change detections, such as land cover change and tiny object changes like vehicles.

### 4.1.2. Experiment Setup

For our experiment, we used two VHR-CD models as our baselines. The first is the 16-channel SNUnet [22]. This model effectively detects changes using a complex encoding mechanism and dense skip-connections. The second is Siam-ResUnet, which uses traditional Siam-Conc as its baseline and Resnet50 as its encoder. Both models were pre-trained on all datasets we used. We also resized all images in the datasets to 256 × 256 pixels to make it easier for the generator part within the framework to be trained. This resizing also ensures that the resolution of the synthetic images generated is consistent with that of the target domain images. Meanwhile, the generator, the target model and other trainable modules are jointly trained on the target domain for 150 epochs with a batch size 32.

Our framework was developed using PyTorch and was deployed on two NVIDIA RTX 3090 GPUs, each corresponding to a framework composed of a single Baseline model. We used the AdamW optimiser during the training of the framework, with an initial learning rate of 0.005 and weight decay of 0.01. We implemented two learning rate adjustment strategies to handle potential anomalies during individual training sessions. The first 30 epochs used the Linear Learning Rate scheduling strategy, with an adjustment rate initialised to $1 \times 10^{-6}$. Subsequent epochs used the Polynomial Learning Rate scheduling strategy, with the power set to 1.0.

### 4.1.3. Evaluation Metrics

We employed F1-score, precision, recall, and changed intersection over union (cIoU) to evaluate the framework performance. Precision shows how many positively predicted cases were True Positive (TP); Recall indicates how many TPs were correctly identified. The F1-Score is the harmonic mean of precision and recall. It's beneficial when the class distribution is imbalanced. cIoU refers to the Intersection over Union (IoU) for the change class. In change detection tasks, change areas usually occupy a smaller proportion, and calculating the IoU for unchanged areas followed by computing the mean IoU (mIoU) can significantly diminish the apparent differences in model performance. This approach fails to reveal the actual performance disparities of models in change detection tasks. All evaluation metrics are calculated as follows:

$$\text{F1-score} = 2 \times \frac{(\text{Precision} \times \text{Recall})}{(\text{Precision} + \text{Recall})} \tag{19}$$

$$\text{Precision} = \frac{\text{True Positives}}{(\text{True Positives} + \text{False Positives})} \tag{20}$$

$$\text{Recall} = \frac{\text{True Positives}}{(\text{True Positives} + \text{False Negatives})} \tag{21}$$

$$\text{cIoU} = \frac{\text{True Positives}}{(\text{True Positives} + \text{False Positives} + \text{False Negatives})} \tag{22}$$

Furthermore, we mark the FNs, i.e., the missed detection areas, in red, and the FPs, i.e., the erroneous detection areas, in blue, to enable clearer qualitative results.

### 4.2. Experiment Results

Figures 6 and 7 demonstrate the qualitative visualisation results of cross-regional and cross-scenario adaptation, whereas Tables 2 and 3 display the quantitative evaluation metrics for cross-regional and cross-scenario adaptation. Tables 4–6 display the quantitative accuracy for ablation studies, including the effect of modules AAM and ISM, and the combination of loss functions.

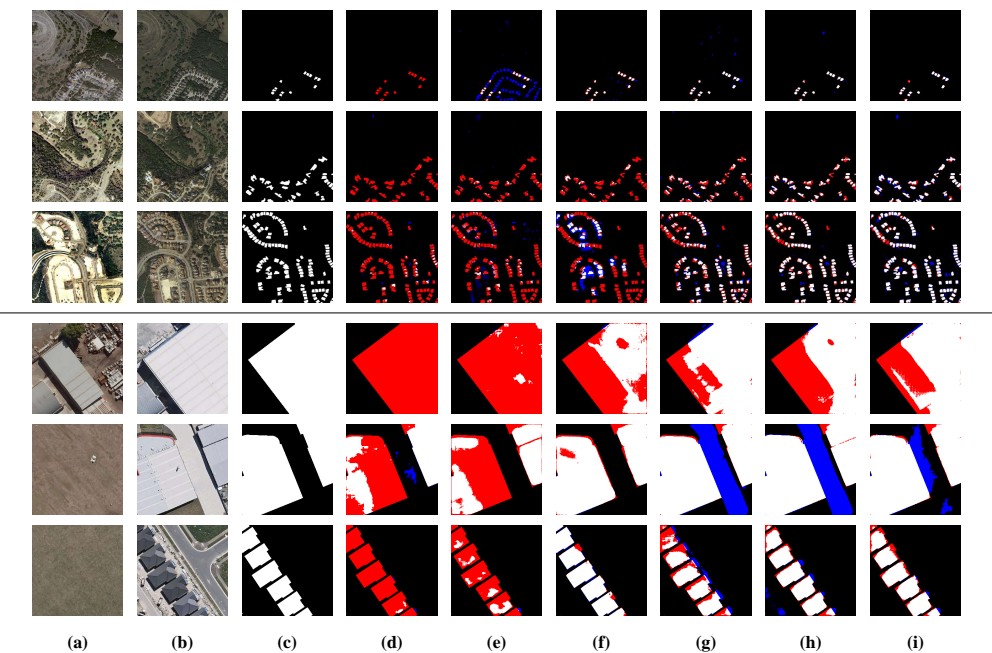

|  (a)  |  (b)  |  (c)  |  (d)  |  (e)  |  (f)  |  (g)  |  (h)  |  (i)  |

**Figure 6.** The qualitative results of Cross-regional adaptation, the upper half of the figure shows WHU-CD → LEVIR-CD, while the lower half shows LEVIR-CD → WHU-CD. (**a**–**c**): Bi-temporal VHR image pairs and change labels; (**d**): Source model-only; (**e**): ColourMapGAN; (**f**): CGDA-CD; (**g**): SGDA; (**h**): Ours (Siam-ResUnet); (**i**): Ours (SNUnet).

**Table 2.** Comparison Evaluation Metrics of Cross-Region.

| Method | Source Free | WHU-CD → LEVIR-CD | | | | LEVIR-CD → WHU-CD | | | |
|---|---|---|---|---|---|---|---|---|---|
| | | cIoU | F1 | Precision | Recall | cIoU | F1 | Precision | Recall |
| Source model-only | ✗ | 24.82% | 30.18% | 56.47% | 20.59% | 26.61% | 31.22% | 58.54% | 21.29% |
| ColourMapGAN | ✗ | 54.97% | 68.38% | **78.78%** | 60.41% | 52.51% | 63.37% | 72.32% | **56.39%** |
| CGDA-CD | ✗ | 54.38% | 68.75% | 68.42% | **69.09%** | 52.33% | 62.83% | 71.19% | 56.22% |
| SGDA | ✗ | 55.52% | 69.49% | 78.00% | 62.65% | 53.01% | 63.30% | 73.39% | 55.65% |
| Ours(Siam-ResUnet) | ✓ | 55.96% | 70.83% | 77.57% | 65.17% | 53.18% | **63.79%** | **74.41%** | 55.82% |
| Ours(SNUnet) | ✓ | **56.01%** | **71.14%** | 76.25% | 66.68% | **53.77%** | 63.76% | 73.72% | 56.17% |

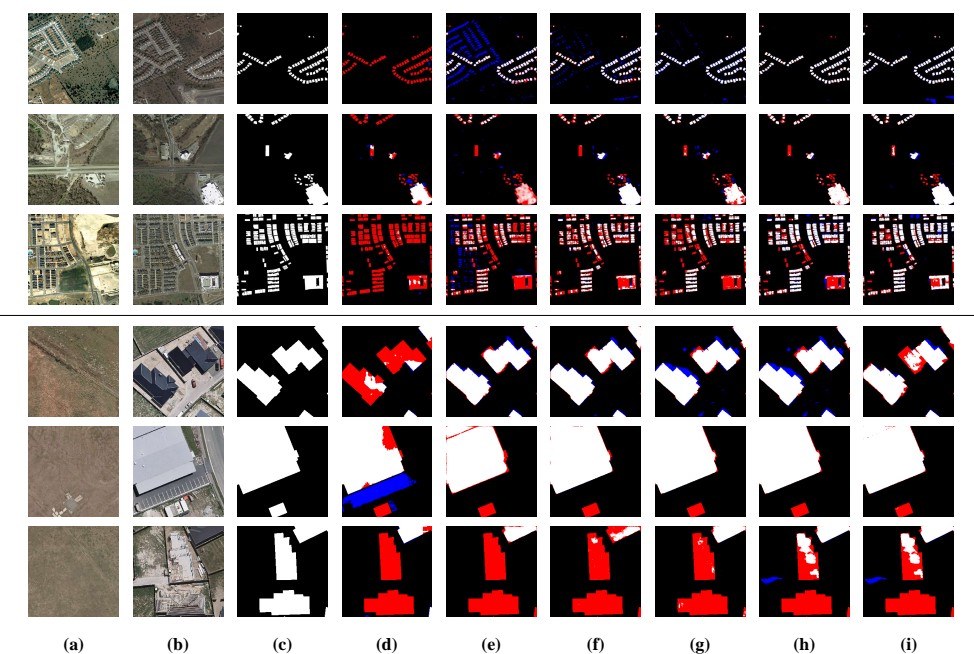

**Figure 7.** The qualitative results of Cross-Scenario adaptation, the upper half of the figure shows CDD → LEVIR-CD, while the lower half shows CDD → WHU-CD. (**a–c**): Bi-temporal VHR image pairs and change labels; (**d**): Source model-only; (**e**): ColourMapGAN; (**f**): CGDA-CD; (**g**): SGDA; (**h**): Ours(Siam-ResUnet); (**i**): Ours(SNUnet).

**Table 3.** Comparison Evaluation Metrics of Cross-Scenario.

| Method | Source Free | CDD (Season-Varying) → LEVIR-CD | | | | CDD (Season-Varying) → WHU-CD | | | |
|---|---|---|---|---|---|---|---|---|---|
| | | cIoU | F1 | Precision | Recall | cIoU | F1 | Precision | Recall |
| Source-only | ✗ | 20.02% | 27.34% | 55.51% | 18.14% | 14.31% | 22.53% | 49.50% | 14.58% |
| ColourMapGAN | ✗ | 50.17% | 59.77% | 73.33% | 50.44% | 46.89% | 52.91% | 59.11% | 47.88% |
| CGDA-CD | ✗ | 51.68% | 63.30% | 62.17% | **64.47%** | 32.14% | 40.40% | 53.17% | 32.58% |
| SGDA | ✗ | 51.48% | 62.52% | 64.48% | 60.68% | 49.85% | 57.17% | 66.47% | 50.15% |
| Ours(Siam-ResUnet) | ✓ | 52.16% | 63.67% | **73.84%** | 55.97% | 51.54% | 59.52% | 68.41% | **52.67%** |
| Ours(SNUnet) | ✓ | **52.88%** | **64.37%** | 72.78% | 57.71% | **52.01%** | **60.93%** | **72.77%** | 52.41% |

**Table 4.** Ablation Evaluation Metrics of Cross-Region.

| Method | CDD (Season-Varying) → LEVIR-CD | | | | CDD (Season-Varying) → WHU-CD | | | |
|---|---|---|---|---|---|---|---|---|
| | cIoU | F1 | Precision | Recall | cIoU | F1 | Precision | Recall |
| Ours(w/o AAM & ISM) | 28.21% | 31.66% | 54.71% | 22.28% | 26.92% | 29.48% | 48.15% | 21.24% |
| Ours(w/o ISM) | 40.16% | 46.78% | 52.54% | 42.16% | 45.57% | 50.51% | 61.68% | 42.77% |
| Ours(w/o AAM ) | 49.85% | 61.96% | 66.19% | 58.24% | 49.10% | 59.66% | **75.56%** | 49.29% |
| Ours | **56.01%** | **71.14%** | **76.25%** | **66.68%** | **53.77%** | **63.76%** | 73.72% | **56.17%** |

**Table 5.** Ablation Evaluation Metrics of Cross-Scenario.

| Method | CDD (Season-Varying) → LEVIR-CD | | | | CDD (Season-Varying) → WHU-CD | | | |
|---|---|---|---|---|---|---|---|---|
| | cIoU | F1 | Precision | Recall | cIoU | F1 | Precision | Recall |
| Ours(w/o AAM & ISM) | 28.74% | 32.08% | 42.07% | 25.93% | 29.17% | 35.14% | 45.18% | 28.75% |
| Ours(w/o ISM) | 39.69% | 42.00% | 51.24% | 35.58% | 36.28% | 38.61% | 48.45% | 32.09% |
| Ours(w/o AAM ) | 45.15% | 49.38% | 59.67% | 42.12% | 42.34% | 46.63% | 57.41% | 39.26% |
| Ours | **52.88%** | **64.37%** | **72.78%** | **57.71%** | **52.01%** | **60.93%** | **72.77%** | **52.41%** |

**Table 6.** cIoU of Different Loss Function Combination.

| Loss Function | | | | Cross-Region | | Cross-Scenario | |
|---|---|---|---|---|---|---|---|
| $\mathcal{L}_{TAR}$ | $\mathcal{L}_{BNS}$ | $\mathcal{L}_{AAM}$ | $\mathcal{L}_{ISM}$ | WHU-CD → LEVIR-CD | LEVIR-CD → WHU-CD | CDD → LEVIR-CD | CDD → WHU-CD |
| ✓ | ✗ | ✗ | ✗ | 20.47% | 21.44% | 20.28% | 15.32% |
| ✓ | ✓ | ✗ | ✗ | 28.21% | 26.92% | 28.74% | 29.17% |
| ✓ | ✓ | ✓ | ✗ | 40.16% | 45.57% | 39.69% | 36.28% |
| ✓ | ✓ | ✓ | ✓ | **56.01%** | **53.77%** | **52.88%** | **52.01%** |

## 5. Discussion

### 5.1. Comparison

#### 5.1.1. Cross-Regional Adaptation

Figure 6 present the qualitative results of LEVIR-CD → WHU-CD and WHU-CD → LEVIR-CD. Even without the support of source data, the proposed framework is competitive compared to three unsupervised domain adaptation methods with source data, namely ColourMapGAN [31], CGDA-CD [34], and SGDA [32], as shown in the visualised results. It shows that with great accuracy, our framework can effectively recognise the main changes and detect changes in densely distributed and uniformly featured buildings, such as houses. Compared to these methods, our framework reduces false positives to a certain extent, reflected in the reduced number of incorrectly detected change patches in unchanged areas on the result maps. However, details such as edges and textures are inevitably lost. This is evident in the test results on the LEVIR-CD dataset, where significant discrepancies exist between the edges of the change areas and the Ground Truth. On the WHU-CD dataset, this is manifested as hole structures within large-sized change areas. Although these issues also occur in unsupervised methods, the lack of source data posed challenges in recognising details in our framework.

Table 2 shows the quantitative evaluation metrics for LEVIR-CD → WHU-CD and WHU-CD → LEVIR-CD. Regarding evaluation metrics, our framework significantly improves compared to other UDA methods. On the LEVIR-CD dataset, the framework improved Recall while maintaining stable Precision. This indicates that it can effectively generate fake samples containing specific semantic features, adapting these features and knowledge to the target domain for more accurate recognition of target domain data. The performance on the WHU-CD dataset also confirms this view. However, while there is a significant improvement in the F1 score compared to D, the best UDA method, the increase in cIoU is relatively tiny. For WHU-CD → LEVIR-CD, the framework improved by 1.65 in the F1 score but only 0.49 in cIoU. The accuracy metrics for LEVIR-CD → WHU-CD show similar results. Combined with observations of the qualitative results, this confirms that our framework still faces specific difficulties in recognising edges, textures, and other details.

#### 5.1.2. Cross-Scenario Adaptation

Figure 7 shows the qualitative results for CDD → LEVIR-CD and CDD → WHU-CD. The results suggest that our framework is efficient in extracting and adapting semantic information of scenes from a source domain model to a target domain in cross-scenario adaptation tasks. The CDD dataset is a season-varying dataset where seasonal variations in objects or phenomena are intentionally ignored. A source model trained on this dataset is designed to identify changes in manufactured features such as land types and roads. Although the framework accurately identifies building changes on the LEVIR-CD dataset, it sometimes shows misclassification instances, such as roads that have undergone significant changes in the image pair. These are not building changes, but the framework erroneously identifies them as "changes". These features are also present in the test results on the WHU-CD dataset. The framework can accurately identify changes in the WHU-CD dataset as well. Still, it struggles to recognise the tiny edges of buildings that result from image cutting and resizing.

Table 3 presents the quantitative accuracy matrices from CDD to WHU-CD and LEVIR-CD. Our framework performs commendably in cross-scene adaptation tasks. Compared to Method C, which fails to recognise cross-scene change features, our approach maintains relatively high levels of Precision and Recall, achieving the best cIoU. However, in the CDD → LEVIR-CD task, our method does not perform as well regarding Recall. This may be due to overly cautious discrimination by the framework, leading to more False Negatives. The visual results corroborate this, as misjudgments at the edges and building interiors contribute to the decline in Recall. A similar situation is observed in the CDD → WHU-CD task, though it is less pronounced than the former. Overall, we find that the accuracy of cross-scene transfer tasks is lower than that of cross-region adaptation tasks. This is due to significant differences between the source and target datasets regarding resolution, changed features, and data distribution in cross-scene adaptation tasks.

*5.2. Ablation*

A series of ablation experiments were conducted to verify the enhancement of framework performance by the Attention Adaptation Module (AAM) and Intra-domain Self-supervision Module (ISM). These experiments included scenarios without AAM and ISM, with only AAM, with only ISM, and with the complete framework. Moreover, we also discussed the effects of each loss function. The results of these ablation experiments are presented in Tables 4–6.

5.2.1. Effects of AAM

Based on the evaluation metrics, it is evident that the framework's performance significantly declines in the absence of AAM. This is particularly noticeable in the decrease in Recall, which implies that the framework identifies a smaller proportion of 'truly changed areas'. This indicates a diminished focus on genuinely changed areas. Additionally, the framework is confounded in recognising changed areas due to inherent noise features in the data. The primary aim of proposing AAM was to concentrate on the differences between change features in the source and target domains. By enabling the target model to focus on the knowledge about change features from the source domain, it can effectively align the target domain's intrinsic change features with those of the source domain, thereby enhancing the detection performance of changed areas. The observed reduction in accuracy across all experiments corroborates this viewpoint.

5.2.2. Effects of ISM

The evaluation metrics indicate a significant performance gap between the framework without ISM and the one that includes ISM. This gap is particularly noticeable in cross-scenario adaptation tasks, where the inclusion of ISM leads to a qualitative improvement in the cIoU for both tasks. ISM is a self-supervised module that aims to facilitate the target model's self-supervised training by generating highly credible pseudo-labels. The quality of these pseudo-labels directly affects the performance of the framework. Without them, the target model lacks an effective supervision mechanism to constrain its inference results. The pseudo-labels provided by ISM can address this issue to a certain extent. Similarly, in cross-domain adaptation tasks, ISM significantly enhances the framework's accuracy in detecting changes in building areas.

5.2.3. Effects of Loss Functions

In Table 6, we can see how different loss functions affect the framework's effectiveness. When only using $\mathcal{L}_{TAR}$, the necessary loss function for the target model in the change detection task, the framework's performance is similar to the results obtained by directly inferring with the source model on the target data. This is generally not enough to accomplish the transfer task. However, when we introduce $\mathcal{L}_{BNS}$, the framework's accuracy improves slightly. Nevertheless, the enhancement brought about by BNS's features and constraints on the cIoU metric is almost negligible in practical change maps. On the other

hand, including $\mathcal{L}_{AAM}$ considerably enhances the framework's accuracy. This shows that the attention mechanism can effectively constrain the feature distribution between models and minimise the differences as much as possible. Furthermore, with the inclusion of $\mathcal{L}_{ISM}$, the pseudo-label generation process is restrained, allowing the target model to use more accurate pseudo-labels in the computation of $\mathcal{L}_{TAR}$ and directly improving the framework's performance in this self-supervised task.

## 6. Conclusions

This paper introduces a source-free UDA framework for VHR Image change detection tasks to address challenges such as the unavailability of source data, lack of annotated target data and inability to perform conventional training and transfer tasks. This framework proposes a GAN generator-based domain generation method for knowledge transfer and enhances the efficiency of capturing pixel-level change features by a dual-attention mechanism. Furthermore, we utilise an intra-domain self-supervised module to generate more correct difference maps as pseudo-labels in the target domain, thereby maximising the extraction of practical knowledge. Through extensive comparative experiments and ablation studies, we have confirmed the effectiveness of this framework in various scenarios and demonstrated its competitive performance to existing source-driven UDA methods, in cross-regional tasks, it has 0.6% cIoU and 1.5% F1 score up, and in cross-scenario tasks has 1.4% cIoU and 1.9% F1 score up. Meanwhile the qualitative results demostrated that this framework can detect changed area more precisely and avoid much issue like inner hole and error detection. However, it should be noted that the current framework has limitations in accurately identifying the edge information of the change areas. Additionally, it needs help sufficiently concentrating on the target domain change scenarios when dealing with cross-scenario adaptation tasks. In our future work plans, we aim to improve the change detection quality of our framework, mainly on edge accuracy. We will enrich the feature extraction and adaptation process by multi-level feature hierarchy alignment to achieve this. This approach allows the target model to gradually learn features similar or identical to the source model's at multiple adaptation levels, resulting in improved target model performance.

**Author Contributions:** Conceptualization, data curation, formal analysis, investigation, methodology, software, validation, visualization, writing—original draft draft, J.W. Supervision, investigation, funding acquisition, resources, writing—review and editing, C.W. All authors have read and agreed to the published version of the manuscript.

**Funding:** This work was supported in part by the National Key Research and Development Program of China under Grant 2022YFB3903300, and in part by the National Natural Science Foundation of China under Grant T2122014. The numerical calculations in this paper have been done on the supercomputing system in the Supercomputing Center of Wuhan University.

**Data Availability Statement:** Data are contained within the article.

**Conflicts of Interest:** The authors declare no conflict of interest.

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
