# Peer review of "SFDA-CD: A Source-Free Unsupervised Domain Adaptation for VHR Image Change Detection"

_remotesensing, doi:10.3390/rs16071274_

Round 1
Reviewer 1 Report
Comments and Suggestions for Authors
Author Response
Dear Editors and Reviewers:
Thank you for reviewing the manuscript titled "SFDA-CD: A Source-Free Unsupervised Domain Adaptation for VHR Image Change Detection" with the manuscript ID remotesensing-2921359. Your feedback is extremely valuable to us and has significantly contributed to improving the quality of the manuscript. We have carefully reviewed and considered your comments, and based on the issues you raised, we have made necessary revisions and corrections to the manuscript.
The response to your comments is attached.

Reviewer 2 Report
Comments and Suggestions for Authors
The paper entitled "SFDA-CD: A Source-Free Unsupervised Domain Adaptation for VHR Image Change Detection" presents a methodology for adapting a change detection model on very high resolution images to be used with other information sources without having to generate new labeled datasets for the new information sources.
The proposal is relevant since the main problem of supervised methods is that they require large and expensive training datasets to generate and this method avoids this effort.
The paper is well structured and can be followed in an affordable way.
Some minor recommendations:
Review the bibliographic references; numerous references to conference contributions do not provide dates or the place where they were developed.
In Algorithm 1, SFDA-CD could be improved with a text describing what it does. For example, the meaning of N, which constrains the loop.
Also avoid that the figure captions have the same caption as in Figures 3 and 4: The illustration of AAM.
The experiments section identifies the dataset used, the model used and how they have implemented and experimented with it. No reference is made in the article as to whether the code that defines the set of neural networks and the scripts that drive the methodology will be accessible in a code repository that will make it possible for future contributions to make comparisons.
Finally, we request that in the abstract, contributions and conclusions the authors include quantitative values of the improvement of their methodological proposal compared to the models with which it is compared, in addition to the aforementioned advantage of not having to label the data.
Author Response
Dear Editors and Reviewers:
Thank you for reviewing the manuscript titled "SFDA-CD: A Source-Free Unsupervised Domain Adaptation for VHR Image Change Detection" with the manuscript ID remotesensing-2921359. Your feedback is extremely valuable to us and has significantly contributed to improving the quality of the manuscript. We have carefully reviewed and considered your comments, and based on the issues you raised, we have made necessary revisions and corrections to the manuscript.
The response to your comments has been attached.

Reviewer 3 Report
Comments and Suggestions for Authors
The authors' original achievement is the synthesis of the unsupervised domain adaptation platform for detecting VHR image changes.
1. The manuscript in its current form is a research report, not a scientific article.
2. In the Introduction section, there is no formulation of the scientific thesis and the objectives of the work, which become its chapters.
3. The Conclusion section, as it stands, is mostly a summary of it. Here, quantitative and qualitative research results should be presented synthetically.
4. The manuscript should be edited so that the Conclusions section is an extension of the Abstract, without the need to read the entire text of the work.
5. In the Conclusions section, present the plan for further research on the topic of paper.
Author Response

(The authors gave the same response as above.)

Round 2
Reviewer 1 Report
Comments and Suggestions for Authors
The authors of the article provided comprehensive answers to the questions raised and added corresponding content in the revised version. At present, all parts of the article are relatively complete, and there are no obvious errors in the overall logic and methods. Note that there are several punctuation and capitalization errors in the article, and you can correct them before final publishment.